# OceanGPT: A Large Language Model for Ocean Science Tasks

## Abstract

Ocean science, which delves into the oceans that are reservoirs of life and biodiversity, is of geart significance given that oceans cover over 70% of our planet's surface. Recently, advances in Large Language Models (LLMs) have transformed the paradigm in science. Despite the notable success in other domains, current LLMs often fall short in catering to the needs of domain experts like oceanographers, and the potential of LLMs for ocean science is under-explored. The intrinsic reason is the immense and intricate nature of ocean data as well as the necessity for higher granularity and richness in knowledge. To alleviate these issues, we introduce **OceanGPT**, the first-ever LLM in the ocean domain, which is expert in various ocean science tasks. We propose DoInstruct, a novel framework to automatically obtain a large volume of ocean science instruction data, which generates instructions based on multi-agent collaboration. Additionally, we construct the first oceanography benchmark, **OceanBench**, to evaluate the capabilities of LLMs in the ocean domain. Though comprehensive experiments, **OceanGPT** not only shows a higher level of knowledge expertise for oceans science tasks but also gains preliminary embodied intelligence capabilities in ocean technology[1].

## 1 Introduction

Ocean science, which delves into the intricacies of oceans that cover over 70% of our planet's surface, is essential not only for understanding the rich reservoirs of life and biodiversity but also for recognizing their pivotal role in regulating the global climate and supporting economies. Recently, advances in Large Language Models (LLMs) (OpenAI, 2023; Chowdhery et al., 2022) have transformed the paradigm in science domains such as medical science (Moor et al., 2023), molecular science (Fang et al., 2023), protein science (Lin et al., 2023) and geoscience (Deng et al., 2023). However, the potential for the LLM in ocean science is under-explored.

Despite remarkable success in general domain, current LLMs still do not fully meet the specific demand of oceanographers. This inadequacy is primarily due to: (1) The immense volume and intricate nature of ocean data. As ocean science research progresses, acquiring ocean data becomes increasingly challenging, which makes enhancing the oceanic understanding both a golden opportunity and a significant hurdle. (2) The necessity for higher granularity and richness in knowledge. The data requirements faced by researchers are becoming increasingly intricate and diverse. Ocean science encompasses various domains and subjects, each with its distinct data attributes and patterns.

To alleviate these issues, we introduce **OceanGPT**, the first-ever LLM in the ocean domain, which is expert in various ocean science tasks. Specifically, we propose DoInstruct, an efficient ocean science instruction generation framework that capitalizes on multi-agent collaboration. Each agent in our designed framework is considered an expert in a specific domain (Topic) and is responsible for generating the corresponding data. For the advancement of ocean science research using LLMs, we also create a benchmark called **OceanBench** to evaluate the capabilities in ocean science tasks. Our benchmark data is based on criteria manually evaluated by ocean experts, and can accurately reflect the capabilities that LLMs possess in the field of ocean science.

Through extensive experiments, **OceanGPT** shows superiority for diverse ocean science tasks. As depicted in Figure 1, our model can comprehensively answer questions according to the instructions

---

[1]Codes are in the supplementary materials and will be released.

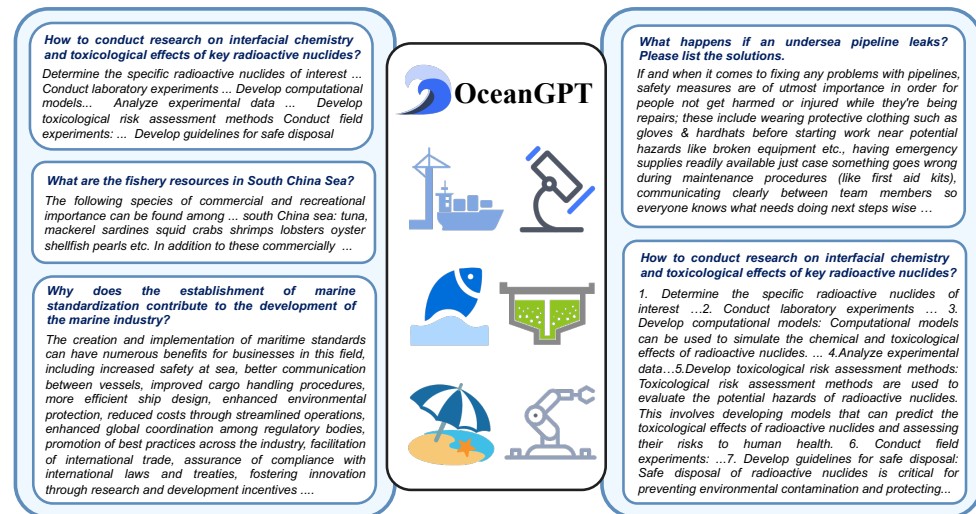

Figure 1: Capabilities of **OCEANGPT**.

of oceanographers, which demonstrates its expertise in oceanography. We also explore the potential of **OCEANGPT** from the perspectives of ocean science and ocean engineering. Specifically, we integrate ocean robotics instructions into the training data and evaluate the ability of **OCEANGPT** via code or console commands. **OCEANGPT** not only demonstrates a higher level of knowledge expertise but also gains preliminary embodied intelligence capabilities in ocean technology.

Our contributions can be summarized as follows:

- We introduce **OCEANGPT**, the first ocean LLM, which shows superiority for various ocean science tasks. It can answer oceanographic questions according to the instructions of oceanographers, demonstrating expertise in oceanography.

- We propose DOINSTRUCT, an automated domain instruction evolving framework that constructs the ocean instruction dataset by multi-agent collaboration. Our framework effectively alleviates the difficulty of obtaining ocean domain data.

- Extensive experiments demonstrate the superiority of **OCEANGPT** in our build **OCEAN-BENCH**. **OCEANGPT** not only demonstrates a higher level of knowledge expertise for oceans science tasks but also gains preliminary embodied intelligence capabilities.

## 2 RELATED WORK

**Large Language Models.** The landscape of LLM (Brown et al., 2020; Chowdhery et al., 2022; Touvron et al., 2023a;b) has rapidly evolved and achieved a series breakthroughs. Rae et al. (2021); Zhang et al. (2022); Thoppilan et al. (2022); Scao et al. (2022); Zeng et al. (2023) have explored the performance across a wide range of model scales and broadened the application scope. LLMs also exhibit capabilities in various aspects, including reasoning and knowledge interaction (Qiao et al., 2023a; Zhang et al., 2023a; Qiao et al., 2023b; Wang et al., 2023a; Xi et al., 2023). To build LLMs, instruction tuning (Wei et al., 2022; Zhang et al., 2023b; Ouyang et al., 2022) is a crucial technique to alignment with user preferences and desired outputs. Taori et al. (2023) uses self-instruct (Wang et al., 2023d) to generate data from manually instructions. Chiang et al. (2023); Xu et al. (2023) uses user-shared conversations and AI-generated data for tuning. Different from those, we introduce an automatic and effective domain instruction generation framework via multi-agent collaboration.

**Science Large Language Models.** LLMs have emerged as cornerstone models in addressing challenges within scientific research. Singhal et al. (2022) explores the potential of clinical LLMs and introduces a human evaluation framework and instruction prompt tuning. Moor et al. (2023) proposes generalist medical AI that is capable of handling diverse medical tasks using self-supervised

learning on large datasets. Kraljevic et al. (2021) introduces MedGPT, a model using EHR data and Named Entity Recognition tools for predicting future medical events. BioGPT (Luo et al., 2022) is a transformer language model pre-trained on biomedical literature for improved text generation and mining. Theodoris et al. (2023) describes Geneformer, a model pre-trained on single-cell transcriptomes for making predictions with limited data in network biology. Lin et al. (2023) demonstrates the prediction of atomic-level protein structure from primary sequences using scaled-up language models. Deng et al. (2023) introduces the first LLM specifically designed for geoscience, including its training and benchmarking protocols. Different from previous works, we are the first LLM for ocean science tasks and explore its potential for ocean research.

## 3 OCEANGPT

To obtain **OCEANGPT**, we first construct the training corpus for ocean science and pre-train a large model in ocean science based on LLaMA-2 (Touvron et al. (2023b) in Section 3.1. Then we propose DOINSTRUCT, an automated framework for domain instruction generation to build a ocean domain-specific instruction dataset. Our framework leverages multi-agent collaboration and utilizes ocean literature to automatically generate a large volume of domain-specific instructions for ocean science tasks (Section 3.2). The overview of our **OCEANGPT** is shown in Figure 2.

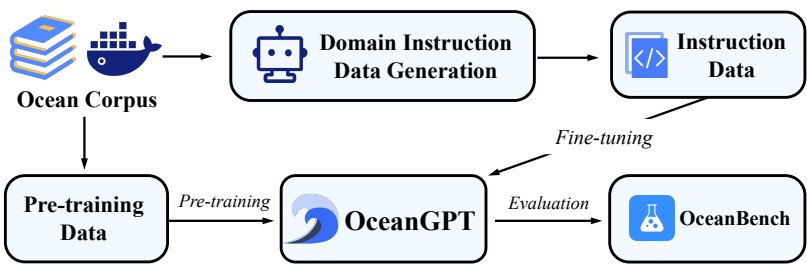

Figure 2: Overall framework of training **OCEANGPT**.

### 3.1 PRE-TRAINING STAGE

To pre-train the foundation model for ocean science tasks, it is essential to construct the pre-training dataset specific to ocean science. Therefore, we first collect a raw corpus of 67,633 documents from **open-access literature**. Specifically, we utilize the Python package *pdfminer* to convert the content of literature files into plain text. To ensure the quality and consistency of the data, further processing of the collected dataset is necessary. We apply regular expressions to filter out figures, tables, headers, footers, page numbers, URLs and references. Additionally, any extra spaces, line breaks, and other non-text characters are removed. The processed documents cover various aspects of ocean science such as ocean physics, ocean chemistry, ocean biology, geology, hydrology, etc. It is important to note that special characters, emoticons, and garbled characters are also replaced or eliminated during this process. We also employ *hash-based methods* to de-duplicate the data, which helps reduce the risk of over-fitting in the model and enhances its generalization capability. The detailed statistics is shown in Table 1 (Appendix A.3).

### 3.2 DOMAIN INSTRUCTION DATA GENERATION

As ocean science research deepens, researchers are facing increasingly complex and diversified data demands. Ocean science corpus contains multiple fields and topics, and each topic has its unique data characteristics and patterns. To effectively simulate and generate this data, we propose a domain instruction generation framework DOINSTRUCT to obtain ocean instructions $H$ by multi-agent collaboration. Each agent is considered an expert in a **specific domain (topic)** and is responsible for generating the corresponding data. It not only ensures the professionalism and accuracy of the data but also allows for the parallel and efficient generation of a large amount of data. Note that framework also has greater flexibility, allowing us to independently optimize and adjust for different domains (e.g., astronomy).

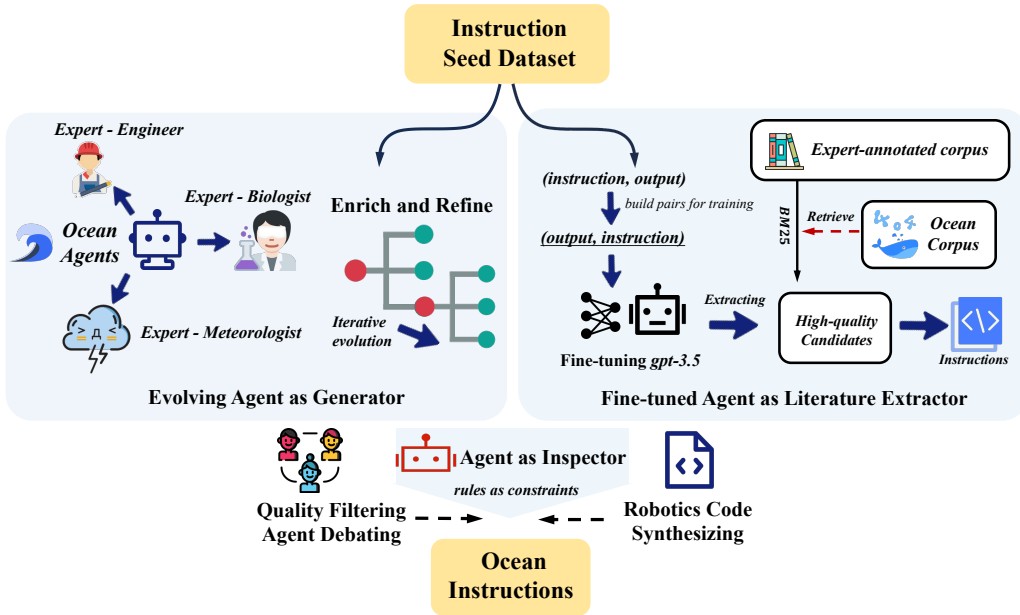

Figure 3: Procedure of our proposed DOINSTRUCT. We use agents as experts for each ocean topic and make them rapidly expand the instructions by collaboration. We design three agent roles: **evolving generator**, **fine-tuned literature extractor** and **inspector with rule constraints**.

**Ocean Topic Definition.** To provide researchers with a clear and organized resources, we manually categorize the data in ocean science into five major ocean topics, which are based on the expertise of experts in oceanography. The definitions of these five topics comprehensively cover all the main areas of ocean science and are relatively independent. The detailed explanation for the five major topics is described as follows:

- *Science and research* focuses on the fundamental scientific theories and research related to the ocean, such as ocean currents, sea temperatures and ocean biodiversity. This portion of data separately helps drive the advancement of pure scientific research and theories.

- *Technology and engineering* encompasses aspects ranging from ocean measurements, observational equipment, and ship engineering to ocean energy development. Such categorization aids in a more focused exploration of ocean engineering and technological needs, while also facilitating interdisciplinary research with other engineering disciplines.

- *Resources and development* includes fisheries, minerals, oil and gas, as well as other sustainable development resources. It is set for a better examination and planning of the rational development of ocean resources.

- *Ecology and environment.* Environmental protection and ecological sustainability are currently global hot topics. It helps to address issues such as ocean pollution, ecological degradation, and the impact of climate change on the oceans in a more focused manner.

- *Life, culture and others.* The ocean is not only a natural resource or a subject of scientific research; it is also an integral part of culture and lifestyle. This category consists of aspects ranging from history and culture to the mutual influences between the ocean and human societal activities, such as tourism, leisure, art, faith, and more.

While these five topics are distinct, there might be some overlap as well. For instance, some issues related to ocean environmental protection might also be associated with the technology of ocean engineering. For the sake of convenience in data analysis, in the actual construction of the dataset, we map each sample to the most relevant category it belongs to.

---

**Algorithm 1** Domain Instruction Data Generation

---

**Require:**
- Seed dataset $S$ with format (instruction, output);
- Ocean literature corpus $O$;
- Pre-defined rules $R$ for filtering;

**Ensure:** High-quality instruction dataset $H$

Initialize empty datasets: Step1Data $= \emptyset$, Step2Data $= \emptyset$, $H = \emptyset$

                                  ▷ Agent Collaboration as Domain Experts;

**for** each sample in $S$ **do**
    instruction, output ← sample
    enriched_sample ← Agent_Enrich(instruction, output)
    refined_sample ← Agent_Refine(instruction, output)
    Step1Data ← Step1Data $\cup$ refined_sample
**end for**

                                  ▷ Fine-Tuned Agent as Literature Extractor;

RetrievedTexts ← BM25_Retrieve(O)
Model $M$ ← Agent_FineTuning(S)
**for** each document in RetrievedTexts **do**
    output ← document.content
    instruction ← Model(output)
    Step2Data ← Step2Data $\cup$ (instruction, output)
**end for**

                                  ▷ Agent as Inspector with Rule Constraints;

MergedData ← Agent_Inspector(Step1Data, Step2Data, R)

                                  ▷ Quality Control by Debating;

**for** each sample in MergedData **do**
    instruction, output ← sample
    debate_result ← Agent_Debate(instruction, output)
    **if** debate_result is high-quality **then**
        $H$ ← $H \cup$ sample
    **end if**
**end for**
**return** $H$

---

**Agents as Domain (Ocean) Experts** In Figure 3, we leverage agents as domain experts for each ocean topic and make them rapidly expand the instructions by collaboration. We first collect the seed instruction data and then propose three strategies by using multiple agents acting as experts. The seed instruction dataset comes from annotations by ocean science researchers, which includes 5 major categories, over 500 sub-categories, and a total of more than 10,000 data entries.

- **Evolving Agent as the Generator**. We design an evolving approach that selects samples from the seed instruction dataset and simultaneously calls upon two agents (*gpt-3.5-turbo*) to evolve the selected samples. The evolution procedure includes two aspects: (1) we enrich the content of the sample by having the agent automatically add relevant background knowledge to it; (2) we guide the agent to refine the sample by conducting a more in-depth analysis of specific concepts or entities. Through multiple rounds of iterative execution, our method can rapidly expand the existing seed dataset, which allows for the rapid expansion of both the breadth and depth of information.

- **Fine-Tuned Agent as the Literature Extractor**. As shown in Figure 3, we collect a smaller expert-annotated corpus and use the *BM25* to retrieve high quality sentences in a larger ocean dataset. We use the retrieved texts are used as high-quality candidate samples. Meanwhile, we fine-tune *gpt-3.5-turbo* in the seed instruction dataset. Thus, we can regard the fine-tuned agent as the literature extractor. In other words, it can automatically ask questions (*instruction*) from the unsupervised ocean science corpus (*output*). Therefore, we use the agent to automatically build pairs of *(instruction, output)* on external ocean science literature.

- **Agent as the Inspector with Rule Constraints**. For the massively generated instructions, we use the pre-defined rules as constraints and perform filtering on the data. These rules include syntactic and semantic constraints as well as some basic definitions in the ocean domain. We describe these rules using natural language because many constraints and norms related to ocean science cannot

be directly represented with expressions. Therefore, we provide prompts to the *gpt-3.5-turbo* API as demonstrations, letting it play the role of an Inspector. Our approach ensures that the generated ocean instruction data is of higher quality. The detailed prompt agent is shown in Appendix A.5.

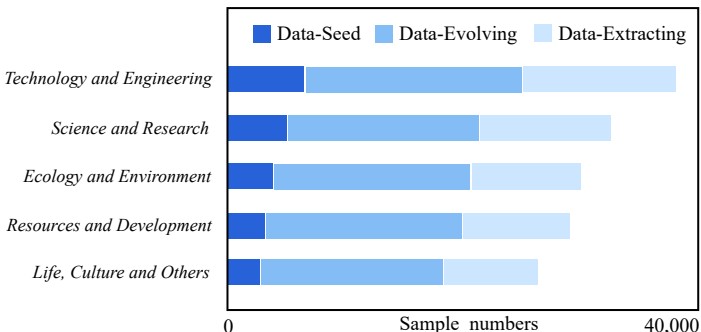

Figure 4: Statistics of our final instruction dataset. We use DOINSTRUCT to expand more than 15,000 instructions (*data-evolving*, *data-extracting*) from the seed data (*data-seed*).

Finally, we assign two extra *gpt-3.5-turbo* agents as roles to debate the quality of data and obtain high-quality instruction dataset. The designed framework is capable of rapidly constructing a ocean science dataset using multi-agents, and by incorporating external knowledge, it overcomes the limitations inherent to the agents themselves. Our framework can also be effectively applied to the construction of directive data in other scientific domains. It should be noted that we separately synthesize robot instructions to equip **OCEANGPT** with the capability to interact with the environment. The statistics of our dataset are shown in Figure 4 and the procedure is described in Algorithm 1.

**Quality Control for the Dateset.** We ask domain experts to carefully review and check data to ensure quality. Specifically, the human volunteers with no experience are first trained to make sure they have a comprehensive understanding of the task. Then we develop a platform that can help experts to sample instances from the instruction dataset. Next, the trained domain experts are asked to validate if there are potential errors in the sampled instances. The final IAA (inter-annotator agreement) score for our dataset is 0.82, which satisfies the research purpose. We will continue to maintain the benchmark by adding new data and tasks.

## 4 BENCHMARKING OCEAN SCIENCE TASKS

We provide detailed explanations of the experimental setup and the baseline models in Section 4.1. In Section 4.2, we construct an ocean-specific benchmark **OCEANBENCH** to evaluate the capabilities of our **OCEANGPT**. We describe the *GPT4*-based evaluation metrics in Section 4.2.

### 4.1 IMPLEMENTATION DETAILS AND BASELINES

For the pre-training stage, we pre-train our **OCEANGPT** based on the LLaMA-2 (Touvron et al., 2023b) model for seven days with six A800 Nvidia GPUs. For the instruction-tuning stage, we employ the LoRA method (Hu et al., 2021) to fine-tune the pre-trained model and choose three baseline models for comparison. We use the chat version of LLaMA-2 (*Llama-2-7b-chat-hf*[2]) , which is a generative text model optimized for dialogue use cases. *Vicuna-1.5*[3] (Chiang et al., 2023) is a chat assistant which fine-tune LLaMA-2 on dataset collected from ShareGPT. *ChatGLM2-6B*[4] is the optimized version of ChatGLM2 (Zeng et al., 2023). The detailed experimental settings are shown in Table 3 (Appendix A.4).

---

[2] https://huggingface.co/meta-llama/Llama-2-7b-chat-hf
[3] https://huggingface.co/lmsys/vicuna-7b-v1.5
[4] https://huggingface.co/THUDM/chatglm2-6b

## 4.2 Datasets and Metrics

**OceanBench.**    To evaluate the capabilities of LLMs for oceanography tasks, we design a benchmark called **OceanBench**. Our benchmark includes a total of 15 ocean-related task types, such as question-answering, extraction, and description tasks. Our evaluation samples are automatically generated from the seed dataset and have undergone deduplication and manual verification. For the quality control, the sampling strategy is adopted. Those disagreed cases are made final decisions by the annotators and bad cases are returned to be fixed. The distribution of our **OceanBench** and the statistics can be found in Appendix A.3.

**Automatic Evaluation.**    To evaluate the performance and reduce reliance on manual evaluation, we leverage GPT-4 as the evaluator for our experimental results. Inspired by Wang et al. (2023c) and Wang et al. (2023b), we utilize an effective calibration method to evaluate the performance of two LLMs. For each testing question, given two responses and from two LLMs, we query the GPT4 to obtain the comparison result. We note that LLMs are sensitive to the position of responses, therefore, alleviating the positional bias is very important. To balance the position bias, we exchange the order of the responses to form the new prompt. The final evaluating result is the sum of the test results for the two prompts with their order swapped.

**Human Evaluation.**    To validate our proposed framework, we also collect the output data in different settings and evaluate it by human evaluation. We employ 5 students with an ocean science background as human annotators. For each evaluation setting, we sample a set of 200 examples and human annotators will rank the outputs they prefer. The total expense is about 500 US dollars.

## 5 Evaluation

### 5.1 Insights from Performance Results

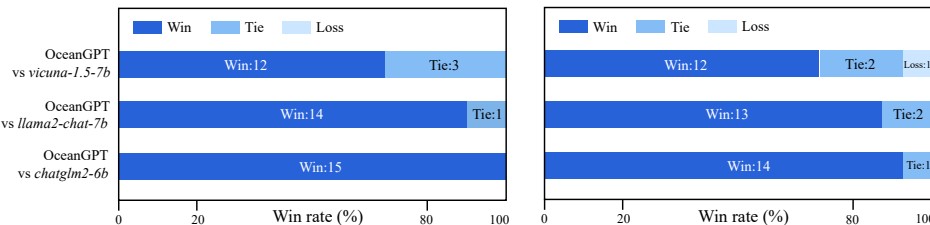

Figure 5: Ocean task-level results. **Left**: Automatic evaluation. **Right**: Human evaluation. Compared to baselines, **OceanGPT** performs better than *llama2-chat-7b*, *vicuna-1.5-7b* and *chatglm2-6b* in both two settings. The instance-level result is in Table 11 (Appendix A.2).

**OceanGPT can obtain better performance than previous open-sourced LLMs.**    In Figure 5, we compare the performance of **OceanGPT** with the three baseline models across 15 subtasks at the task-level in the ocean domain. We utilize both automatic and human evaluators, then compute the *win rate (%)* with baseline models. Compared to the baselines (*llama2-chat-7b*, *vicuna-1.5-7b*, *chatglm2-6b*)[5], **OceanGPT** outperforms in the majority of tasks, which demonstrates the effectiveness of the proposed approach.

**OceanGPT excels in a range of ocean science tasks.**    As shown in Figure 6, we present detailed automatic evaluation experimental results in the **OceanBench**. It can be clearly seen that our model is superior to baseline language models in the vast majority of tasks. Note that previous open-sourced LLMs even fail to handle several expertise ocean tasks (e.g., Editing). While our designed multi-agent data generation framework can effectively act as experts in various subfields of the ocean domain, which indicates that **OceanGPT** is a better expert in various ocean domains.

---

[5]Before the ICLR 2024 deadline, we have only trained OceanGPT-7b, thus we compare open-sourced LLMs around 7B. We are running resources to train OceanGPT-30b, and will compare with more LLMs in the future.

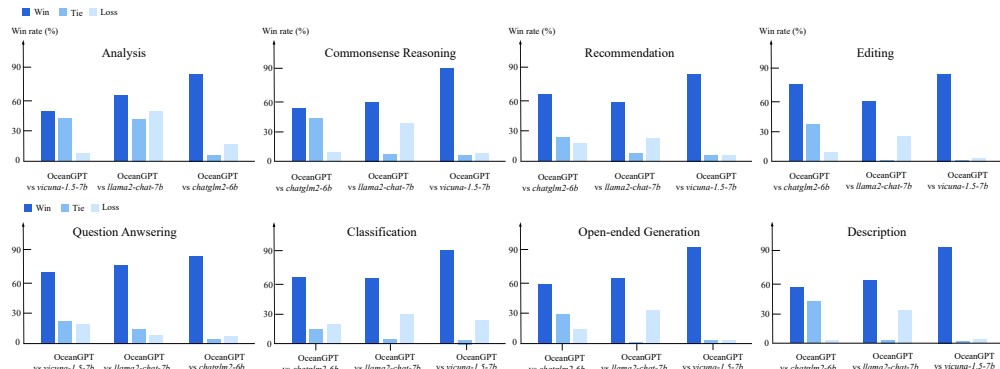

Figure 6: Evaluation results of **OCEANGPT** in the ocean science tasks in **OCEANBENCH**. The complete experimental results are shown in the Table 10 (Appendix A.2).

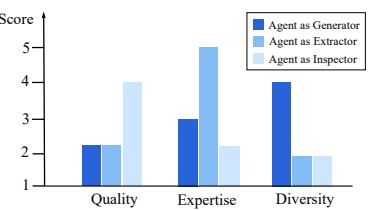

Figure 7: Performance analysis for agents.

**DOINSTRUCT are the effective ocean data generators by multi-agent collaboration.** As shown in Figure 7, we design three indicators to measure the data generation effect of our proposed method from the perspectives of knowledge quality, expertise and diversity. We use manual evaluation to calculate the scores of the three indicators from 1 to 5. The higher the score, the better the effect of the testing model. It can be seen that the evolving generator agent can effectively enhance the richness of ocean data. When the extraction agent is at work, the expertise of the content is greatly improved. At the same time, the inspector agent plays a significant role in enhancing the quality of the generated data. The results show that multi-agent collaboration is effective for ocean instruction generation.

## 5.2 EXPLORING THE POTENTIAL OF OCEANGPT

In this section, we explore the potential of **OCEANGPT** from the perspectives of ocean science and ocean engineering. For ocean science (Section 5.2.1), we focus on the key scientific issues of nuclear pollution in the ocean environment. For ocean engineering (Section 5.2.2), we use the Gazebo [6] as the simulator to test **OCEANGPT**'s ability to control underwater robots.

### 5.2.1 OCEANGPT FOR OCEAN SCIENCE

In Figure 8, given the question *"How to conduct research on interfacial chemistry and toxicological effects of key radioactive nuclides"*, we compare the outputs of our model and *vicuna-1.5-7b*. The result shows that **OCEANGPT** demonstrates a higher level of knowledge expertise when describing the content of radioactive nuclide research. Its textual content is not only clear in structure and well-organized, but also covers various aspects of radioactive nuclide research, from experimental design to data analysis, and then to risk assessment and disposal guidelines. In contrast, although *vicuna-1.5-7b* has clear expression and logicality, it lacks depth and specific content related to radioactive nuclides. Overall, **OCEANGPT** has advantages in terms of knowledge expertise, quality, and richness. The complete outputs of both models are shown in the Table 4 (Appendix A.6).

### 5.2.2 OCEANGPT FOR OCEAN ENGINEERING

Ocean engineering focuses on the design, development, and management of structures and systems within the ocean environment. It plays an indispensable role in harnessing the vast potential of the oceans while ensuring sustainable and secure maritime operations. To facilitate interaction between **OCEANGPT** and the external world, we integrate machine code instructions into its training data.

---

[6]https://github.com/uuvsimulator/uuv_simulator

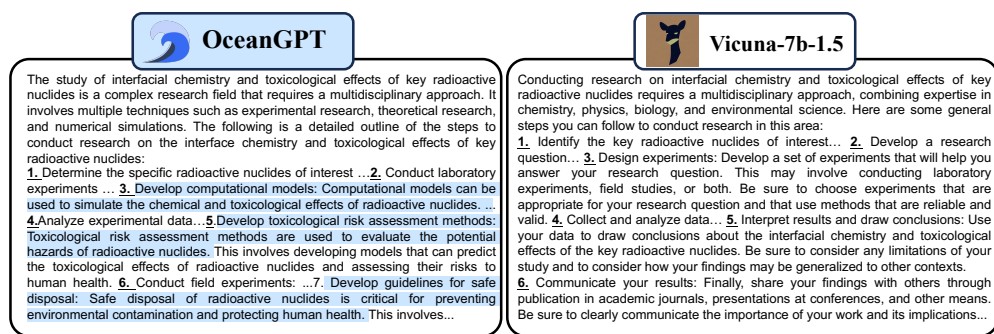

Figure 8: Performance analysis on ocean science issues. We use  blue  font to represent the different texts by **OCEANGPT**. The instruction is:  `How to conduct research on interfacial chemistry and toxicological effects of key radioactive nuclides?`

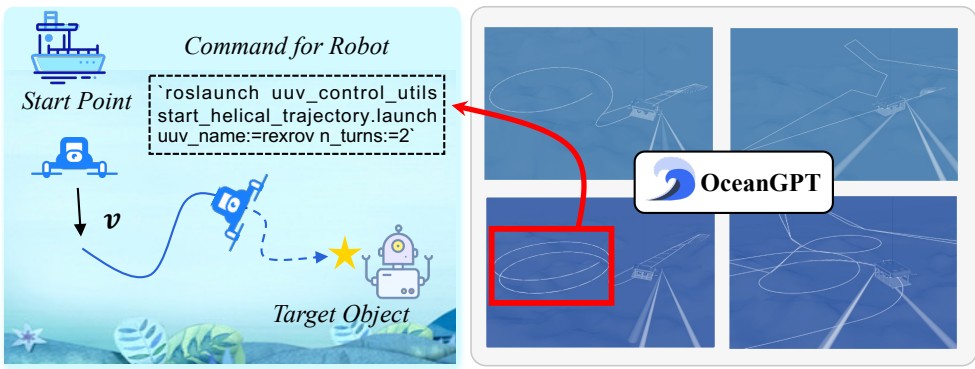

Figure 9: Our model can be instructed for underwater robot control in the simulation platform of Gazebo, which shows **OCEANGPT** gains preliminary embodied intelligence capabilities.

As depicted in Figure 9, **OCEANGPT** can instruct underwater robots via code or console commands, allowing them to execute basic path-finding operations. In this example, by using natural language as a prompt, the **OCEANGPT** can automatically generate code (the robot generate a double helix path) for underwater robot to operate based on human instructions. In fact, the experimental result suggests that **OCEANGPT** has the potential to acquire embodied intelligence. Though we make preliminary attempts for ocean robot interaction, it paves the way for future advanced oceanic models to undertake intricate robotic control and planning tasks.

## 6  CONCLUSION

In this paper, we introduce **OCEANGPT**, the first-ever oceanographic pre-trained language model, which is expert in various ocean science tasks. To alleviate the difficulties for obtaining ocean data, we propose an domain construction framework called DOINSTRUCT, which constructs the ocean instruction dataset by multi-agent collaboration. Each agent in our designed framework is considered an expert in a specific topic and is responsible for generating the corresponding data. Our generated dataset consists of diverse instructions to align the desired behaviors in ocean science issues. Additionally, we establish the first oceanography benchmark, **OCEANBENCH**, to evaluate the capabilities of LLMs in ocean science. Though comprehensive analysis in ocean science and ocean engineering, **OCEANGPT** not only demonstrates a higher level of knowledge expertise for oceans science tasks but also gains preliminary embodied intelligence capabilities. We will continue to improve **OCEANGPT** by training larger models (e.g., 30b, 70b) and maintain **OCEANBENCH** by adding new data and tasks.

ETHICAL CONSIDERATIONS

The datasets and model in this paper are designed for exploratory analysis. Within the domain of LLMs, the distribution of both pre-training data and instructional data might exhibit biases, potentially influencing the model's outputs. Throughout experiments, **we have carefully examined the sampled data, eliminating any content that could be deemed toxic or offensive**. Yet, it's possible that **OCEANGPT** might occasionally yield incorrect or imprecise results. Hence, users leveraging the insights provided by this model should exercise their discretion and bear the risk.

REPRODUCIBILITY STATEMENT

**Codes, datasest are in the supplementary materials and will be released on GitHub, OCEANGPT will be released on Huggingface**. We provide technical details of in Appendix A.4 and prompt design in Appendix A.5.

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

# A    APPENDIX

## A.1    LIMITATIONS AND ETHICAL CONCERNS

### A.1.1    BIAS IN DATA DISTRIBUTION

In the realm of large language models, the distribution of pre-training data and instruction data can be subject to substantial biases, which can shape the outputs of these models. Pre-training data for language models often comes from the internet, a vast and potentially biased source of information. The Internet content is inherently skewed, reflecting the biases of its contributors, and hence may not represent a balanced global perspective. Similarly, instruction data can also carry the biases of the humans who create these instructions. For instance, instruction developed by individuals with a particular cultural, socioeconomic, or educational background may inadvertently favor specific perspectives, languages, or communication styles and marginalize others. This bias in data distribution can result in models that reinforce existing prejudices, lack cultural sensitivity, or fail to accurately understand and generate content in underrepresented languages or dialects.

### A.1.2    HALLUCINATION IN LARGE LANGUAGE MODELS

Although LLMs have shown tremendous success in general domains of NLP, there is a notable issue regarding their tendency to produce hallucinations. Hallucinations refer to instances where LLMs occasionally generate content that deviates from the user's input, contradicts previously generated context, or conflicts with established world knowledge. By developing strategies to address the issue of hallucination, LLMs can better align their outputs with user intent, preserve coherence within generated content, and enhance their overall utility in real-world applications.

## A.2    COMPLETE RESULTS IN OCEANBENCH

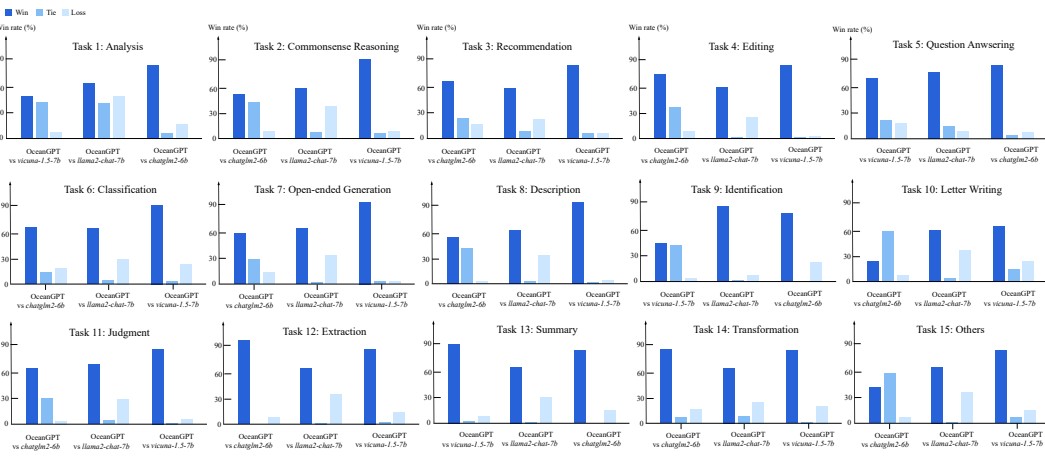

Figure 10:    Automatic evaluation results of **OCEANGPT** in all ocean science tasks in **OCEAN-BENCH**.

## A.3    DATA STATISTICS AND ANALYSIS

## A.4    IMPLEMENTATION DETAILS

## A.5    PROMPT DESIGN IN THIS WORK

## A.6    DETAILED RESULT FOR OCEAN SCIENCE ISSUES

| Journal | Num | Journal | Num |
|---|---|---|---|
| Oceanography | 1,359 | Ocean Science | 1,168 |
| Ocean Dynamics | 2,825 | Marine Biology | 10,485 |
| Aquatic Sciences | 2,522 | Fisheries Science | 2,371 |
| Maritime Studies | 313 | Marine Biotechnology | 1,794 |
| Ocean Science Journal | 717 | Physical Oceanography | 455 |
| Marine Geophysical Research | 1,155 | Journal of Oceanography | 2,580 |
| Journal of Physical Oceanography | 3,588 | Marine Ecology Progress Series | 1,494 |
| Fish Physiology and Biochemistry | 3,251 | Atmospheric and Oceanic Optics | 1,156 |
| Reviews in Fish Biology and Fisheries | 1,158 | Journal of Oceanology and Limnology | 3,510 |
| Journal of Atmospheric and Oceanic Technology | 3,298 | Journal of Ocean Engineering and Marine Energy | 266 |
| Journal of the Marine Biological Association of the United Kingdom | 7,676 | Journal of Offshore Mechanics and Arctic Engineering | 2,015 |

Table 1: The statistics of the pre-training corpus. **Num** denotes the number of documents.

| Task | Num | Task | Num | Task | Num |
|---|---|---|---|---|---|
| Analysis | 674 | Classification | 895 | Judgment | 655 |
| Commonsense Reasoning | 1024 | Open-ended Generation | 930 | Extraction | 1078 |
| Recommendation | 1,089 | Description | 1,246 | Summary | 149 |
| Editing | 1,075 | Identification | 464 | Transformation | 401 |
| Question Answering | 1,230 | Letter Writing | 359 | Others | 157 |

Table 2: The statistics of **OCEANBENCH**. **Num** denotes the number of samples.

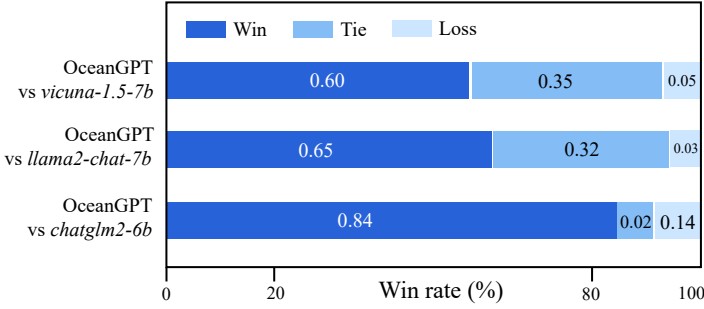

Figure 11: Instance-level results (automatic evaluation) in 15 tasks of **OCEANBENCH**.

| Hyperparameter | |
|---|---|
| Fine-tuning method | LoRA |
| Batch Size | 512 |
| Device† | NVIDIA A800 |
| GPU number | 6 |
| Learning Rate (LR) | $1e-4$ |
| LoRA $r$ | 8 |
| LoRA $\alpha$ | 16 |
| LoRA Dropout | 0.05 |
| Epoch | 10 |

Table 3: Detailed experimental settings.

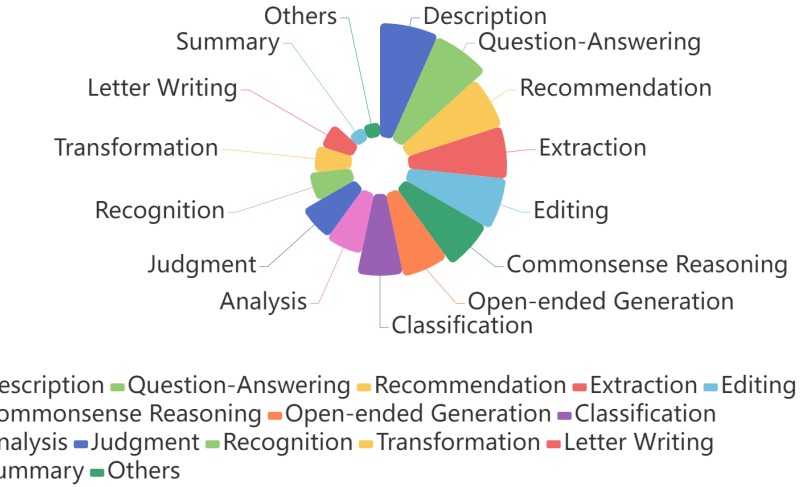

Figure 12: Distribution of our OCEANBENCH.

---

**Prompt for "Fine-Tuned Agent as the Literature Extractor":**
You are a helpful ocean assistant. You are to extract the question from each of the answer provided.
Answer: This is a seahorse, belonging to the family Syngnathidae. Seahorses are vertebrates commonly found in tropical and subtropical waters. They have unique morphology and biological characteristics and are important organisms in marine ecosystems.

---

**Prompt for "Evolving Agent as the Generator":**
Assuming you are an expert in marine engineering and resources, please keep the meaning of the following sentences unchanged and provide as much professional knowledge as possible.
Sentences:Please recommend some mineral resources found in the East China Sea.

---

**Prompt for "Agent as the Inspector with Rule Constraints":**
Assuming you are an inspector in marine science, please filter and judge the sentences in 'Sentences' based on the constraints provided below:
Constraints: Keyword Filter: Focus on literature that prominently mentions the terms 'coral reefs', 'ocean acidification', or 'deep-sea exploration'. Date Range: Only consider articles published between 2010 and 2022. Author Filter: Prioritize works by the Oceanic Research Institute. Type of Literature: Specifically look for 'experimental studies' and 'review articles'. Exclude 'conference papers'. Geographical Focus: Highlight research that pertains to the Pacific Ocean region. Language Constraint: Only select literature written in English. Abstract Inclusion: Ensure the abstract contains the phrase 'climate impact'. Abstract Exclusion: Exclude any literature whose abstract mentions 'laboratory simulation'.

---

**Prompt for automatic evaluation using GPT4:**
Please check if following sentences contain rich ocean related information. If so, output "related". Otherwise, output "unrelated".
Sentences: Dissolved organic carbon (DOC) represents the largest pool of reduced carbon in oceans and plays important roles in the ocean carbon cycle and food webs . DOC comprises nearly half of the riverine organic carbon flux into oceans. Riverine DOC is involved in numerous ecosystem functions, including key roles in chemical and biological processes. Refractory and labile DOC are, respectively, important for carbon sequestra-tion in the ocean and a vital food source for marine bacteria.

---

Figure 13: The prompt example that we use in this work.

| **Instruction:** How to conduct research on interfacial chemistry and toxicological effects of key radioactive nuclides? | |
|---|---|
| **OCEANGPT** | The study of interfacial chemistry and toxicological effects of key radioactive nuclides is a complex research field that requires a multidisciplinary approach. It involves multiple techniques such as experimental research, theoretical research, and numerical simulations. The following is a detailed outline of the steps to conduct research on the interface chemistry and toxicological effects of key radioactive nuclides: 1. Determine the specific radioactive nuclides of interest: Different radioactive nuclides have different chemical properties and toxicological effects. Therefore, it is necessary to select specific radioactive nuclides of interest and establish a research plan. 2. Conduct laboratory experiments: Laboratory experiments are crucial for studying the chemical properties and toxicological effects of radioactive nuclides. The experiments involve measuring the chemical properties of radioactive nuclides, such as their reactivity, stability, and chemical reactivity. 3. Develop computational models: Computational models can be used to simulate the chemical and toxicological effects of radioactive nuclides. This involves developing mathematical models and algorithms to predict the behavior of radioactive nuclides under different conditions. 4. Analyze experimental data: Experimental data obtained from laboratory experiments need to be analyzed and interpreted. This involves studying the experimental results and comparing them with theoretical predictions. 5. Develop toxicological risk assessment methods: Toxicological risk assessment methods are used to evaluate the potential hazards of radioactive nuclides. This involves developing models that can predict the toxicological effects of radioactive nuclides and assessing their risks to human health. 6. Conduct field experiments: Field experiments are necessary to study the behavior of radioactive nuclides in real-world environments. This involves measuring the chemical and toxicological effects of radioactive nuclides in different ecosystems and studying their distribution and transport. 7. Develop guidelines for safe disposal: Safe disposal of radioactive nuclides is critical for preventing environmental contamination and protecting human health. This involves developing guidelines for the safe disposal of radioactive nuclides and their storage and transport. |
| *vicuna-1.5-7b* | Conducting research on interfacial chemistry and toxicological effects of key radioactive nuclides requires a multidisciplinary approach, combining expertise in chemistry, physics, biology, and environmental science. Here are some general steps you can follow to conduct research in this area: 1. Identify the key radioactive nuclides of interest: Start by identifying the specific radioactive nuclides that you want to study. This may include elements such as uranium, plutonium, and cesium, among others. 2. Develop a research question: Once you have identified the key radioactive nuclides, develop a research question that addresses an important aspect of their interfacial chemistry and toxicological effects. This question should be specific and focused, and should guide your research efforts. 3. Design experiments: Develop a set of experiments that will help you answer your research question. This may involve conducting laboratory experiments, field studies, or both. Be sure to choose experiments that are appropriate for your research question and that use methods that are reliable and valid. 4. Collect and analyze data: Once you have conducted your experiments, collect and analyze the data you have generated. Be sure to carefully document your methods and results, and to interpret your data in a clear and concise manner. 5. Interpret results and draw conclusions: Use your data to draw conclusions about the interfacial chemistry and toxicological effects of the key radioactive nuclides. Be sure to consider any limitations of your study and to consider how your findings may be generalized to other contexts. 6.Communicate your results: Finally, share your findings with others through publication in academic journals, presentations at conferences, and other means. Be sure to clearly communicate the importance of your work and its implications for interfacial chemistry and toxicology. |

Table 4: Performance analysis on ocean science issues. The input prompt instruction is `How to conduct research on interfacial chemistry and toxicological effects of key radioactive nuclides?`

