# OpenReview forum: "OceanGPT:  A Large Language Model for Ocean  Science Tasks"
_ICLR.cc/2024/Conference — ICLR 2024 Conference Withdrawn Submission_

### Official Review · Reviewer_5Gh6 · 2023-10-27

**Soundness:** 1 poor
**Presentation:** 3 good
**Contribution:** 1 poor
**Rating:** 3
**Confidence:** 5

**Summary:**

This paper provides a fine-tuned version of LLaMa 2, OceanGPT, which is meant to be specifically used for ocean studies and related tasks. It was also finetuned by a series of GPT 3.5 derived agents, allowing it to carry out a variety of tasks, such as providing instructions for underwater robots and extracting relevant parts of literature.

**Strengths:**

- The authors aim to train a model specifically for an under-served field of study, that of oceanography
- They create a benchmark for testing AI models capabilities on ocean-adjacent tasks
- They create a corpus of open-access literature about ocean studies.

**Weaknesses:**

- The 'instruction seed generation' approach described in section 3.2. is fundamentally flawed because it utilizes other LLMs to generate this data, meaning that it is liable to contain hallucinations and not be reliable from a scientific perspective
- The evaluation carried out has several significant shortcomings (see my questions below) - notably that there is a lack of details regarding how it was carried out, or how reliable GPT4 is as an evaluator.
- The insistence of the authors on "embodied intelligence" is not proven or described in detail, and the whole section about the instructions for underwater robots is unclear to me - how this testing was carried out, how it was evaluated, etc. (see questions below)

**Questions:**

- Do you check licenses of content that you use?
- "The seed instruction dataset comes from annotations by ocean science researchers" - which researchers? what was the annotation procedure?
- It's unclear to me: you use 'gpt-3.5-turbo' to enrich training samples? what about hallucination issues? (in general, all the agents that you use will have this issue, so for me all the data gathered during this step (described in Section 3.2.) is unverifiable and therefore cannot be trusted
- You say that you "leverage GPT-4 as the evaluator for our experimental results" - that's an unacceptable form of evaluation, because GPT-4 is lacking the specific domain knowledge you seek to represent, and because it suffers from the hallucination issues that all LLMs suffer from.
- When you compare models and say that "the result shows that OCEANGPT demonstrates a higher level of knowledge expertise when describing the content of radioactive nuclide research.", how do you measure this?
- You state that "the experimental result suggests that OCEANGPT has the potential to acquire embodied intelligence." - without providing any proof of formal evaluation of the model's ability to write instruction code for underwater robots. You need to develop a procedure for evaluating this before you can make such statements.

---

> ### Author Response · Authors · 2023-11-21
>
> We sincerely thank you for your insightful feedback. Below are our detailed responses to your concerns:
>
> **Q1: Hallucinations and not be reliable from a scientific perspective**
>
> Thanks for your comment. It is a very important issue. In fact, the issue of hallucinations in large scientific models cannot be completely avoided. In our paper, we use LLMs to generate data in order to address the issue of difficult access to ocean-specific data. Meanwhile, we also use a fine-tuned agent to automate the extraction of knowledge from text, specifically to mitigate such hallucinations. In future research, we will continue to focus on this issue.
>
> **Q2: Licenses of content that you use**
>
> During the process of corpus collection, we only use literature that was accessible and meets the requirements.
>
> **Q3: Collecting annotation of the seed dataset**
>
> For the construction of the seed dataset, we employ annotators with rich backgrounds in marine science to assist in building the dataset. Each annotator is responsible for several topics, and they first manually write some high-quality data. Then, we use LLMs to mimic this existing data to generate a large number of similar samples. All these samples are ultimately manually screened and modified by the annotators
>
> **Q4:  The use of evaluator in our work**
>
> Thanks for your suggestions. We use GPT-4 primarily because it is well-suited for comparing the quality of content generated by two language models. However, to avoid the drawbacks (such as hallucination or lack of domain knowledge) of automated evaluation, we also employed manual assessment methods.
>
> **Q5: The measure for the content of radioactive nuclide research**
>
> In fact, when analyzing the content generated by language models, we consider three aspects: quality, expertise, and diversity (as shown in Figure 7). Additionally, in Figure 8, we have marked the expertise demonstrated by OceanGPT in blue.
>
> **Q6: The embodied intelligence for OceanGPT**
>
> Thanks for your comment. Regarding our model for controlling robots, in the paper we have provided proof (in the Gazebo platform) that it is capable of interacting with underwater robots. We will continue to focus on this issue and explore the potential of OceanGPT.

---

### Official Review · Reviewer_bjnA · 2023-10-30

**Soundness:** 3 good
**Presentation:** 3 good
**Contribution:** 2 fair
**Rating:** 5
**Confidence:** 4

**Summary:**

This paper builds a large language model (LLM) for ocean science tasks, namely OceanGPT, which is the first attempt of concerning LLM with ocean science. OceanGPT is firstly pre-trained on the collected open-access literature of ocean corpus based on the LLaMA-2 model,  then fine-tuned on the instruction data generated by the proposed DoInstruct framework based on multi-agent collaboration and five specific ocean topics, and lastly evaluated on the constructed ocean-specific benchmark OceanBench.

**Strengths:**

- It is the first attempt of building a large language model (LLM) for ocean science, which is helpful for the related study.
- The proposed DoInstruct seems flexible for the LLM model on other fields to generate instruction data.

**Weaknesses:**

Major concerns:
- Although the authors made a great effort on building the OceanGPT, the key contributions seem not strong. Most of the operations on building OceanGPT are general for current LLM models. The authors claim that the DoInstruct is novel, but actually the multi-agent collaboration has already been concerned in the field of LLM, for example, the papers collected in the URL of https://github.com/AGI-Edgerunners/LLM-Agents-Papers, and the authors didn't discuss the essential contributions different from them.
- The Appendix seems not finished without any content (text) from A.2 to A.6.

Minor concerns:
- The authors are suggested to pay attention to the writing like typos and grammar errors, for example, the sentence in Page 5: "We use the retrieved texts are used as high-quality candidate samples".

**Questions:**

- What are the contributions that can consider that this work is meaningful to the LLM community?

---

> ### Author Response · Authors · 2023-11-21
>
> We sincerely thank you for your insightful feedback. Below are our detailed responses to your concerns:
>
> **Q1: Discussion about multi-agent collaboration**
>
> Thanks for your constructive suggestion. As you say, the multi-agent collaboration actually has already been concerned in the field of LLMs. Different from those work in the general domain, we introduce the (science) domain instruction generation framework via multi-agent collaboration.
>
> **Q2: Mistakes in the Appendix**
>
> Sorry for our mistakes in the Appendix and we have corrected all the points.
>
> **Q3: Typos and grammar errors**
>
> Sorry for the mistakes and we have corrected it.
>
> **Q4: The contribution to the LLM community**
>
> Thanks for your advice. In fact, we believe that progress of foundational LLMs and innovative applications in science domains are also very import. OceanGPT is the first-ever LLM in the ocean domain. It not only demonstrates a higher level of knowledge expertise for oceans science tasks but also gains preliminary embodied intelligence capabilities.

---

### Official Review · Reviewer_E237 · 2023-11-02

**Soundness:** 2 fair
**Presentation:** 2 fair
**Contribution:** 3 good
**Rating:** 5
**Confidence:** 4

**Summary:**

This study introduces a large language model for ocean science tasks to explore the potential of LLMs for ocean science.

**Strengths:**

This study introduces a large language model for ocean science tasks to explore the potential of LLMs for ocean science. It is a great research topic and beneficial to experts in the ocean science domain.

**Weaknesses:**

While the proposed OceanGPT shows great potential for ocean science tasks, certain details appear to be absent from the manuscript, and valuable information seems to be dispersed throughout the text.

**Questions:**

1.	Additional related work should be included to provide a comprehensive background  and necessity of the proposed work
2.	I recommend including "DoINSTRUCT" in Figure 2.
3.	In Section 3.1, the authors mention that they collected a raw corpus of 67,633 documents. While the source journals are listed in the Appendix, detailed information is missing, such as the criteria for selecting these journals, the specific volumes chosen, and the types of articles included.
4.	In Section 3.2, the authors state that over 10,000 data entries across 500 sub-categories were provided by ocean science researchers. However, they do not explain how these annotations were collected.
5.	In Section 3.2, the introduction to the fine-tuned agent is unclear. What does it mean to "automatically generate questions from the unsupervised ocean science corpus"?
6.	In the title of Figure 4, the authors mention that 15,000 instructions were generated from data seeds. If I understand correctly, they collected 10,000 data seeds. Does this mean that DoINSTRUCT generated an additional 15,000 instructions from the seeds?
7.	In Section 3.2, please provide more details about the quality control steps, such as the number of samples evaluated.
8.	In Section 4, what is meant by "For each testing question, given two responses from two different LLMs"?
9.	In Section 5, could you explain how the win rate is calculated?
10.	The content in the appendix should be reviewed for accuracy and completeness, as there are a few issues:
(1)	Sections A.3, A.4, and A.5 are empty.
(2)	 Some content is missing or cannot be located. For example, in the title of Figure 6, the authors refer to Table 10 in the Appendix, but I was unable to find Table 10.

---

> ### Author Response · Authors · 2023-11-21
>
> We sincerely thank you for your insightful feedback. Below are our detailed responses to your concerns:
>
> **Q1: Additional related work**
>
> Thanks for your advice. We will add more related work in the revised version and provide a comprehensive background.
>
> **Q2: Revision about "DoINSTRUCT" in Figure 2**
>
> Thanks for your suggestion and we will revise the Figure 2.
>
> **Q3: Detailed procedure about documents collection**
>
> For the thematic part, we only select literature related to the field of marine science. For the specific volumes we choose, we prefer to consider publications from recent years to ensure the inclusion of the latest research and developments. At the same time, we will select some historically significant literature to help the model understand the developmental history of the field. Regarding diversity, we choose articles from different publishers and journals to ensure coverage of various research perspectives and methods.
>
> **Q4: The collecting annotations**
>
> For the construction of the seed dataset, we employ dozens of annotators with rich backgrounds in marine science to assist in building the dataset. Each annotator is responsible for several topics, and they first manually write some high-quality data. Then, we use LLMs to mimic this existing data to generate a large number of similar samples. All these samples are ultimately manually screened and modified by the annotators
>
> **Q5: Unclear points about the fine-tuned agent**
>
> Sorry for the confusion. We want to convey that the fine-tuned agent has the ability to extract information from unannotated corpus. We have corrected the mistake in the new version.
>
> **Q6: Instructions generated from data seeds**
>
> Sorry for the mistake here. We collect 10,000 data seeds and our proposed DoINSTRUCT generates an additional 150,000 (not 15,000) samples.
>
> **Q7: Details about the quality control steps**
>
> Sorry for the unclear points here. We randomly sample 10% instances from the generated instruction dataset and ask the trained domain experts to validate if there are potential errors in the sampled instances.
>
> **Q8: The meaning “For each testing question, given two responses from two different LLMs**
>
> Sorry for the confusion here. For each testing question, we query the GPT4 to obtain the comparison result when given two outputs from two LLMs.
>
> **Q9: how the win rate is calculated**
>
> For the task-level calculation, we compare the effectiveness of two models for each task. When one model performs better on the majority of test samples in a single task, it is considered to 'win' that task. For the instance-level computation process, we do not differentiate between specific tasks and instead calculate overall metrics.
>
> **Q10: Incorrect points in the Appendix**
>
> Sorry for our mistakes in the Appendix and we have corrected all the points you mentioned.